# 11,12-EET Regulates PPAR-γ Expression to Modulate TGF-β-Mediated Macrophage Polarization

**DOI:** 10.3390/cells12050700

**Published:** 2023-02-23

**Authors:** Xiaoming Li, Sebastian Kempf, Stefan Günther, Jiong Hu, Ingrid Fleming

**Affiliations:** 1Institute for Vascular Signalling, Centre for Molecular Medicine, Goethe University, 60596 Frankfurt am Main, Germany; 2Max Planck Institute for Heart and Lung Research, Bioinformatics and Deep Sequencing Platform, 61231 Bad Nauheim, Germany; 3Department of Histology and Embryology, School of Basic Medicine, Tongji Medical College, Huazhong University of Science and Technology, Wuhan 430030, China; 4German Center of Cardiovascular Research (DZHK), Partner Site RheinMain, 60596 Frankfurt am Main, Germany

**Keywords:** soluble epoxide hydrolase, PPAR-γ, macrophage, resolution of inflammation, 11,12-epoxyeicosatrienoic acid

## Abstract

Macrophages are highly plastic immune cells that can be reprogrammed to pro-inflammatory or pro-resolving phenotypes by different stimuli and cell microenvironments. This study set out to assess gene expression changes associated with the transforming growth factor (TGF)-β-induced polarization of classically activated macrophages into a pro-resolving phenotype. Genes upregulated by TGF-β included *Pparg*; which encodes the transcription factor peroxisome proliferator-activated receptor (PPAR)-γ, and several PPAR-γ target genes. TGF-β also increased PPAR-γ protein expression via activation of the Alk5 receptor to increase PPAR-γ activity. Preventing PPAR-γ activation markedly impaired macrophage phagocytosis. TGF-β repolarized macrophages from animals lacking the soluble epoxide hydrolase (sEH); however, it responded differently and expressed lower levels of PPAR-γ-regulated genes. The sEH substrate 11,12-epoxyeicosatrienoic acid (EET), which was previously reported to activate PPAR-γ, was elevated in cells from sEH^−/−^ mice. However, 11,12-EET prevented the TGF-β-induced increase in PPAR-γ levels and activity, at least partly by promoting proteasomal degradation of the transcription factor. This mechanism is likely to underlie the impact of 11,12-EET on macrophage activation and the resolution of inflammation.

## 1. Introduction

The recruitment of neutrophils and monocytes to inflamed tissue and their differentiation into macrophages is a crucial step in the inflammatory process. However, once the neutrophil respiratory burst subsides, these and other cells, i.e., macrophages, eosinophils and lymphocytes, need to be removed to restore homeostasis [1]. To support the removal of apoptotic cells and tissue debris (efferocytosis), macrophage function is altered and the cells are reprogramed into a pro-resolving phenotype. Polarized macrophages are frequently broadly classified in two main groups, i.e., classically activated (M1) macrophages which are induced by T-helper 1 (Th-1) cytokines, i.e., the combination of bacterial lipopolysaccharide (LPS) and interferon γ (IFN-γ), and alternatively activated (M2) macrophages that have a pro-resolving and pro-angiogenic phenotype, and are induced by Th-2 cytokines [2,3]. The latter group can be further subdivided into more refined phenotypes: M2a, M2b, M2c, and M2d depending on the use of different stimuli such as interleukin (IL)-4 (M2a) or transforming growth factor β (TGF-β) (M2c). However, the phenotypic characterization of macrophages is highly complicated and there are many more distinct genetic fingerprints and metabolic states than are reflected in a basic M0/M1/M2 classification [4,5,6]. Indeed, additional subtypes have been identified such as macrophages stimulated by oxidized phospholipids, oxidized LDL, or hemoglobin [3].

TGF-β is a master immune regulator and checkpoint that has a major impact on immune suppression within the tumor microenvironment [7]. It has also been implicated in poor responsiveness to cancer immunotherapy [8]. In inflamed tissues, macrophage TGF-β synthesis is stimulated by the uptake of apoptotic cells, a step that is essential for the repolarization of pro-inflammatory macrophages into a pro-resolving phenotype (for reviews see [9,10]). Although endothelial TGF-β signaling drives endothelial-to-mesenchymal transition and vascular inflammation [11], there is some controversy about the exact impact of TGF-β on atherogenesis. Rather than promoting vascular inflammation, there is evidence suggesting that TGF-β signaling plays an important role in the protection against excessive plaque inflammation, loss of collagen content, and induction of regulatory immunity (reviewed by [12,13]). The current study set out to determine changes in macrophage gene expression associated with the repolarization of classically activated (M1) macrophages into a pro-resolving phenotype by TGF-β.

## 2. Materials and Methods

### 2.1. Animals

C57BL/6N mice (6–8 weeks old) were purchased from Charles River (Sulzfeld, Germany). Floxed sEH mice (Ephx2tm1.1Arte) were generated in the C57BL/6N background by TaconicArtemis GmbH (Cologne, Germany) and crossed with Gt(ROSA)26Sortm16(Cre)Arte mice (TaconicArtemis) expressing Cre under the control of the endogenous Gt(ROSA)26Sor promoter to generate mice globally lacking sEH (sEH^−/−^) as described [14]. Age-, gender- and strain-matched mice were used throughout, where possible littermates were used. In cases where studying littermates was not possible, cells were isolated from age-matched C57Bl/6N mice. Preliminary experiments revealed that responses were comparable in cells from C57Bl/6N and Cre-sEH^flox/flox^ mice and different from those of the sEH^−/−^ (Cre+ sEH^flox/flox^) mice. For the isolation of bone marrow, mice were sacrificed using 4% isoflurane in air and cervical dislocation. 

### 2.2. Monocyte Isolation and Macrophage Polarization

Murine monocytes were isolated from the bone marrow of 8–10-week-old mice and differentiated to naïve (M0) macrophages in RPMI 1640 medium (Invitrogen; Darmstadt, Germany), containing 8% heat inactivated FCS supplemented with M-CSF (15 ng/mL, Peprotech, Hamburg, Germany) and GM-CSF (15 ng/mL, Peprotech, Hamburg, Germany) for 7 days. Cells were kept in a humidified incubator at 37 °C containing 5% CO_2_. Thereafter M0 macrophages were polarized to classical activated M1 macrophages by treating with LPS (10 ng/mL; Sigma-Aldrich, Munich, Germany) and IFN-γ (1 ng/mL; Peprotech, Hamburg, Germany) for 12 h. Pro-resolving M2c macrophages were repolarized from M1 macrophages by the addition of TGF-β1 (10 ng/mL; Peprotech, Hamburg) for 48 h, as described [6].

### 2.3. RNA Isolation and Quantitative Real Time PCR (RT-qPCR)

Total RNA was extracted and purified from murine macrophages using Tri Reagent (ThermoFisher Scientific, Karlsruhe, Germany) based on the manufacturer’s instructions. Thereafter, RNA was eluted in nuclease-free water, and its concentration was determined (λ260 nm) using a NanoDrop ND-1000 (ThermoFischer Scientific, Karlsruhe, Germany). For the generation of complementary DNA (cDNA), total RNA (500 ng) was reverse transcribed using SuperScript IV (ThermoFischer Scientific, Karlsruhe, Germany) with random hexamer primers (Promega, Madison, WI, USA). Quantitative PCR was performed using SYBR green master mix (Biozym, Hessisch Oldendorf, Germany) and appropriate primers (Table 1) in a MIC-RUN quantitative PCR system (Bio Molecular Systems, Upper Coomera, Australia). Relative RNA levels were determined using a serial dilution of a positive control. The data are shown relative to the mean of the housekeeping gene 18S RNA.

### 2.4. RNA Sequencing

Total RNA was isolated from macrophages by using RNeasy Micro kit (Qiagen, Hilden, Germany) based on manufacturer’s instructions. The RNA concentrations were determined by using NanoDrop ND-1000 (ThermoFischer Scientific, Karlsruhe, Germany; λ 260 nm). Total RNA (1 µg) was used as input for SMARTer Stranded Total RNA Sample Prep Kit-HI Mammalian (Takara Bio, Kyoto, Japan). Trimmomatic version 0.39 was employed to trim reads after a quality drop below a mean of Q20 in a window of 20 nucleotides and keeping only filtered reads longer than 15 nucleotides [15]. Reads were aligned versus Ensembl mouse genome version mm10 (Ensembl release 101) with STAR 2.7.10a [16]. Aligned reads were filtered to remove: duplicates with Picard 2.25.5 (Picard: A set of tools (in Java) for working with next generation sequencing data in the BAM format), multi-mapping, ribosomal, or mitochondrial reads. Gene counts were established with featureCounts 2.0.2 by aggregating reads overlapping exons on the correct strand excluding those overlapping multiple genes [17]. The raw count matrix was normalized with DESeq2 version 1.30.1 [18]. Contrasts were created with DESeq2 based on the raw count matrix. Genes were classified as significantly differentially expressed at average count >5, multiple testing adjusted *p*-value < 0.05, and log2FC > 0.585 or <−0.585. The Ensemble annotation was enriched with UniProt data [19]. The PCA, volcano plots and pathway enrichment analysis were generated using http://www.bioinformatics.com.cn/srplot, an online platform for data analysis and visualization.

### 2.5. Phagocytosis Assays

M1 polarized macrophages were treated with either solvent or the PPAR-γ antagonist; GW9662 (10 µmol/L, Merck, Darmstadt, Germany), 2 h prior to repolarization to the M2c phenotype using TGF-β1. Thereafter, cells were incubated in RPMI medium supplement with 0.1% BSA (37 °C, 5% CO_2_) and containing pHrodo Red Zymosan bioparticles (10 μg/mL, Invitrogen). After 30 min the cells were washed to remove nonphagocytosed material and zymosan uptake was visualized and quantified using an automated live cell imaging system (IncuCyte, Sartorius, Göttingen, Germany).

### 2.6. PPAR-γ Activity 

PPAR-γ activity was measured using a luciferase construct (PPRE-X3-Luc, Addgene No. 1015) which contains 3 response elements (AGGACAAAGGTCA) upstream of a luciferase reporter [20]. For transfection, M0 macrophages were incubated in RPMI medium containing 0.1% BSA for 2 h prior to the addition of plasmid (100 ng/mL) and Lipofectamin 3000 Transfection Reagent (ThermoFischer Scientific, Karlsruhe, Germany) according to the manufacturer’s instructions. After 24 h, the cells were polarized to M1 and M2c macrophages and stimulated as described in the results section. Luciferase activity was measured 48 h after cell polarization or stimulation with 11,12-EET (1 µmol/L, Cayman Europe, Tallinn, Estonia) using a commercially available kit (ONE-Glo Luciferase Assay System, Promega, Walldorf, Germany).

### 2.7. Immunoblotting

Cells were lysed in RIPA lysis buffer (50 mmol/L Tris/HCL pH 7.5, 150 mmol/L NaCl, 10 mmol/L NaPPi, 20 mmol/L NaF, 1% sodium deoxycholate, 1% Triton and 0.1% SDS) enriched with protease and phosphatase inhibitors and detergent-soluble proteins were resuspended in SDS-PAGE sample buffer. Samples were separated by SDS-PAGE and subjected to Western blotting as described [21]. Membranes were blocked in 3% BSA, incubated with primary antibodies in the blocking solution and horseradish peroxidase-conjugated secondary antibodies. Protein bands were visualized using Lumi-Light plus Western blotting substrate (Roche, Mannheim, Germany) and captured by an image acquisition system (Fusion FX7; Vilber-Lourmat, Torcy, France). The antibody used to identify PPAR-γ was from Santa Cruz (Texas, USA; Cat. # sc-7196, 1:1000), anti-non muscle myosin was from abcam (Berlin, Germany; Cat. # ab75590, 1:1000), and the anti β-actin antibody was from Linaris (Eching, Germany; Cat. # MAK6019, 1:3000). The secondary antibodies were used were: goat anti-rabbit IgG H and L chain specific peroxidase conjugate, and a goat anti-mouse IgG, H and L chain specific peroxidase conjugate (both 1:20,000; Cat. # 401393 and Cat. # 401253, Merck). 

### 2.8. Statistical Analyses

Data are expressed as mean ± SEM. Statistical analysis was performed using Student’s *t* test, or two-way ANOVA with a Tukey’s or Sidak’s post-test. Normalized data were compared using the Kruskal–Wallis rank sum test or Kruskal–Wallis test followed and Dunn’s multiple comparison test (using Prism 9.0.2, GraphPad Software Inc., San Diego, CA, USA) as indicated in the figure legends. Values of *p* < 0.05 were considered statistically significant.

### 2.9. Data and Material Availability

All data associated used this study are present in the paper or the Appendix A.

## 3. Results

### 3.1. Impact of TGF-β-Induced Macrophage Repolarization on Gene Expression

Bone marrow-derived monocytes were isolated from wild-type mice and differentiated to naïve (M0) macrophages in the presence of M-CSF and GM-CSF for 7 days. Thereafter, M0 macrophages were either polarized to classically activated (M1) macrophages by adding lipopolysaccharide (LPS) and interferon (IFN)-γ for 12 h or into pro-resolving M2c by treating M1 macrophages with TGF-β1 for 48 h. RNA-sequencing (RNA-seq) was then performed to identify changes in gene expression associated with macrophage polarization. Principal component analysis (PCA) confirmed that the three groups of macrophages clustered together with clear differences between the polarization types (Figure 1A, Appendix A). As expected, the expression of the classical M1 marker genes *Nos2*, *Ptgs2*, *Il1b*, and *Nlrp3* were significantly higher in M1 versus M2c polarized macrophages. On the other hand, the typical M2/M2c markers, i.e., *Arg1*, *Vegfa* were higher in M2c than in M1 polarized macrophages (Figure 1B). A closer analysis of the genes differentially expressed in M2c versus M1-polarized macrophages revealed additional marked differences, with TGF-β inducing the upregulation of 2952 genes and the downregulation of 2051 genes, including the pro-inflammatory genes *Cxcr4*, *Ptgs2* and *Angptl4*. One of the genes whose expression was significantly increased in M2c macrophages was *Pparg* and gene set enrichment analysis identified changes in the expression of several targets of the peroxisome proliferator-activated receptor (PPAR) family of transcription factors (Figure 1C). PPAR-γ-regulated genes induced by TGF-β included *Angptl4*, *Abcd2*, *Eepd1* and *Tmem8*.

### 3.2. TGF-β-induced M2c Macrophage Polarization Relies on PPAR-γ and Alk5 Activation

To determine the importance of PPAR-γ on the regulation of selected macrophage genes, we determined the impact of the PPAR-γ antagonist GW9662 on the expression of three selected genes in M2c macrophages, i.e., *Cxcr4* (higher in M2c), as well as *Ptgs2* and *Ptx3* (both higher in M1). While there was no significant effect of PPAR-γ antagonism on *Cxcr4* expression, cells treated with GW9662 expressed significantly higher levels of *Ptgs2* and *Ptx3* than cells treated with solvent (Figure 2A). One characteristic of the latter cells is their ability to phagocytose cell debris. While M2c polarized murine macrophages effectively phagocytosed zymosan, particle uptake was clearly reduced in cells treated with the PPAR-γ antagonist (Figure 2B). These observations imply that PPAR-γ activation is required for the down regulation of some pro-inflammatory genes as well as to support the induction of a pro-resolving phenotype by TGF-β.

Consistent with the latter observations, PPAR-γ expression was significantly elevated in M2c versus M1 or M0 macrophages (Figure 3A). Given that M2c polarization was induced by adding TGF-β to M1 polarized macrophages, we determined which TGF-β type I receptor, i.e., activin receptor-like kinase (Alk) 1 or Alk5, mediated the TGF-β-induced increase in PPAR-γ levels. While neither solvent, nor the Alk1 inhibitor; LDN193189 prevented the TGF-β-induced increase in PPAR-γ (Figure 3B), the response was abolished in macrophages pretreated with the Alk5 inhibitor; SD208.

### 3.3. PPARγ Activity in Differentially Polarized Macrophages from Wild-Type and sEH^−/−^ Mice

Next, we set out to determine whether or not mediators known to regulate PPAR-γ were implicated in the TGF-β-induced changes in PPAR levels and gene expression. Given that arachidonic acid metabolism was one of the pathways altered by TGF-β (see Figure 1C), we focused on the role of the potential role of arachidonic acid epoxides. These fatty acid mediators; such as 11,12-epoxyeicosatrienoic acid (11,12-EET), are reported to activate PPAR-γ [22,23,24,25,26], and their cellular levels are largely determined by the activity of the soluble epoxide hydrolase (sEH). Therefore, a luciferase construct containing three PPAR-γ responsive elements was expressed in macrophages from wild-type mice that were then polarized to the M1 and M2c phenotypes. Consistent with the increase in PPAR-γ protein levels, luciferase activity was clearly increased in the M2c macrophages from wild-type mice (Figure 4A). Deletion of the sEH significantly blunted the latter response, which was reflected in the differential expression of PPAR-γ-regulated genes in M2c macrophages from the two genotypes (Figure 4B, Appendix A). Indeed, the well-characterized PPAR-γ-regulated genes *Gipr*, *Vldlr*, and *Rbp1* were all expressed at significantly lower levels in M2c macrophages from sEH^−/−^ versus wild-type mice. A series of fatty acid epoxides are metabolized by the sEH and it was possible to demonstrate higher 11,12-EET and lower levels of its sEH-generated diol; 11,12-dihydroxyeicosatrienoic acid (11,12-DHET), in M2c polarized macrophages from sEH^−/−^ versus wild-type mice (Figure 4C). Moreover, treating M1 polarized macrophages from wild-type mice with 11,12-EET prior to the repolarization with TGF-β, also decreased PPAR-γ activity (Figure 4D).

### 3.4. Regulation of PPAR-γ Levels by 11,12-EET

Comparison of the effects of 11,12-EET versus those of its diol; 11,12-DHET on PPAR-γ protein levels were assessed next. This revealed that the sEH substrate; 11,12-EET, effectively prevented the TGF-β-induced increase in PPAR-γ protein levels in murine macrophages (Figure 5A). 11,12-DHET had no effect. Somewhat unexpectedly, 11,12-EET altered PPAR-γ protein levels without altering *Pparg* expression (Figure 5B) indicating that 11,12-EET may affect the stability of the PPAR-γ protein. At least in adipocytes, ligand-dependent PPAR-γ activation is associated with its subsequent proteasomal degradation [27]. To determine whether or not 11,12-EET decreased PPAR-γ levels by stimulating its proteasomal degradation, experiments were performed in the absence and presence of the proteasome inhibitor MG132. As before, 11,12-EET, but not 11,12-DHET, decreased PPAR-γ protein levels in M2c polarized macrophages and proteasome inhibition prevented the effect (Figure 5C).

## 4. Discussion

The results of this investigation revealed that the TGF-β-dependent repolarization of classically activated (M1) macrophages into a pro-resolving, highly phagocytic phenotype (M2c), relies on the increased expression and activation of PPAR-γ. Deletion of the sEH, to increase cellular levels of fatty acid epoxides, largely prevented TGF-β-induced changes in macrophage gene expression as well as PPAR-γ activation. The effect seen in macrophages from sEH^−/−^ was reproduced in cells from wild-type mice treated with the sEH substrate 11,12-EET and was attributed, at least in part, to the accelerated proteasomal degradation of PPAR-γ.

In our study, we set out to determine changes in macrophage gene expression associated with the repolarization of classically activated (M1) macrophages into a pro-resolving phenotype by TGF-β. It is not surprising that repolarization resulted in marked alterations in macrophage gene expression and a decrease in the expression of pro-inflammatory markers. However, the observation that many of the genes increased in TGF-β-treated macrophages were classical PPAR-γ targets, e.g., *Abcd2*, *Eepd1*, and *Tmem8* was unexpected as TGF-β is a multifunctional cytokine that drives inflammation, fibrosis and cell differentiation, while PPAR-γ activation tends to promote the opposite effects [28]. The impact of TGF-β on gene expression was however consistent with its ability to increase PPAR-γ protein levels as well as transcription factor activity. The changes in gene expression were reflected in functional alterations as zymosan phagocytosis by TGF-β-repolarized macrophages was clearly attenuated in cells treated with a PPAR-γ inhibitor. Our results are consistent with recent reports from other groups that linked the actions of TGF-β with the activation of PPAR-γ signaling (reviewed by [29]). For example, TGF-β signaling and the upregulation of PPAR-γ was reported to be essential for the development and homeostasis of alveolar macrophages [30]. On the other hand, PPAR-γ was reported to interact with Stat3 and Smad3 to interfere with TGF-β signaling and account for the functional antagonism between BMP2 and TGF-β1 pathways in vascular smooth muscle cells [31]. Thus, it seems likely that a complex crosstalk exists between the two pathways. The results of our study also indicate that in macrophages, the TGF-β-induced increase in PPAR-γ expression relies on the activation of Alk5 and as such fits well with a previous report that TGF-β induces M2-like macrophage polarization via Snail-mediated suppression of a pro-inflammatory phenotype, as the induction of Snail is also mediated by Alk5 [20].

PPARs are ligand-inducible transcription factors and are considered important therapeutic targets as they exert anti-atherogenic and anti-inflammatory effects on the vascular wall and immune cells, as well as acting to reduce insulin resistance and dyslipidaemia [32]. However, unlike many receptors that possess a limited number of ligands, there are numerous natural PPAR-γ ligands, in particular mediators derived from polyunsaturated fatty acids [33]. The EETs are among the latter compounds and are generated by the sequential action of cytochrome P450 enzymes and the sEH [34]. These fatty acid mediators are particularly interesting given that their actions have been attributed to PPAR activation [22,23,24,25,26], and the inhibition or deletion of the sEH to increase EET levels has anti-atherosclerotic effects in mouse models [35,36]. In our study, we observed that the activity of PPAR-γ was lower in TGF-β-stimulated macrophages from sEH^−/−^ (EET high) than from wild-type (EET low) mice. While these findings were consistent with the clearly decreased levels of PPAR-γ protein in sEH-deficient macrophages, they seemed to be a direct contradiction of previous reports. The timing of the experiments performed can go a long way to accounting for the observations made as PPAR-γ activity was generally assessed 48 h after TGF-β addition or stimulation with 11,12-EET. Thus, 11,12-EET probably initiates a transient increase in PPAR-γ activity that is terminated by an EET-stimulated pathway that results in PPAR-γ degradation. Given that PPAR-γ levels were not decreased by 11,12-EET in cells treated with MG 132 we propose that 11,12-EET can stimulate the proteasomal degradation of PPAR-γ. Certainly, PPAR-γ levels can be regulated by protein ubiquitination and degradation [27]. Which ubiquitin ligase was activated by 11,12-EET was not studied but there is circumstantial evidence to link 11,12-EET with increased ubiquitination as the cardiomyocyte-specific overexpression of CYP2J2, which generates 11,12-EET and has been reported to promote the ubiquitination of the pattern recognition receptor NLRX1 [37].

Taken together, our results indicate that macrophage levels of the sEH substrate; 11,12-EET, can modulate macrophage polarization by TGF-β, at least partly by promoting the ubiquitination and degradation of PPAR-γ. Given that sEH inhibition prevents the development of atherosclerosis in mice [35,36], and the conversion of inflammatory macrophages to the M2 phenotype drives atherosclerosis regression [38], it may be interesting to determine how much of the phenotype observed can be attributed to changes in PPAR-γ expression.

## Figures and Tables

**Figure 1 cells-12-00700-f001:**
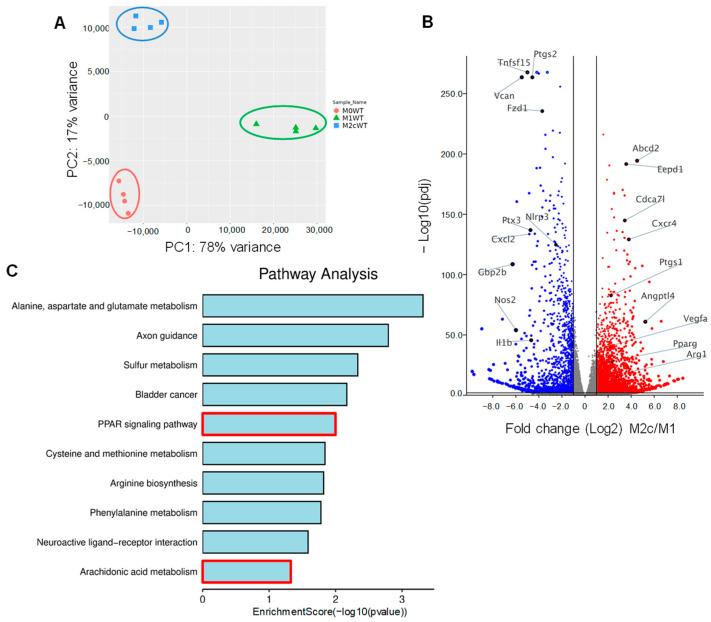
Impact of TGF-β-induced macrophage repolarization on gene expression. (**A**) Principle component analysis showing the clustering of RNA-seq samples from differentially polarized (M0, M1 and M2c) macrophages from wild-type (WT) mice; *n* = 4 mice/group. (**B**) Volcano plot showing differentially expressed genes in M1 and M2c polarized. Blue = genes significantly downregulated and red = genes significantly upregulated in M2c versus M1 macrophages. (**C**) Gene set enrichment analysis of gene expression in M1 and M2c macrophages.

**Figure 2 cells-12-00700-f002:**
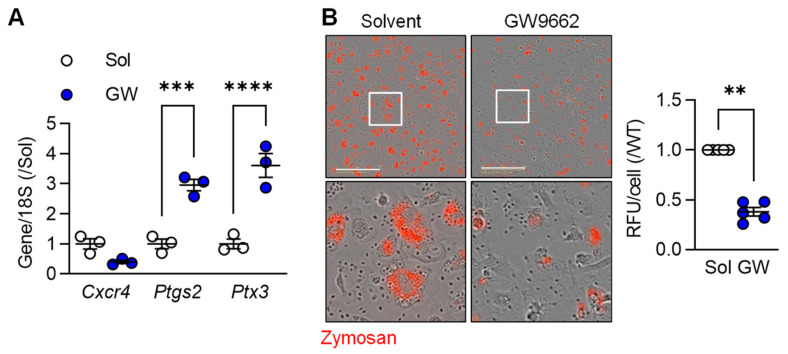
M2c macrophage polarization relies on PPARγ activation. (**A**) Expression of *Cxcr4*, *Ptgs2* and *Ptx3* in M2c polarized macrophages from wild-type mice treated with solvent (0.1% DMSO) or the PPAR-γ antagonist; GW9662 (10 µmol/L) 2 h prior to the addition of TGF-β (*n* = 3 independent experiments, Student’s t test). (**B**) Zymosan phagocytosis by M2c macrophages treated as in A. Images were taken 30 min after zymosan addition and the white boxes indicate the area magnified in the lower panels; bar = 200 µm (*n* = 5/group, Kruskal–Wallis rank sum test). ** *p* < 0.01, *** *p* < 0.001, **** *p* < 0.0001.

**Figure 3 cells-12-00700-f003:**
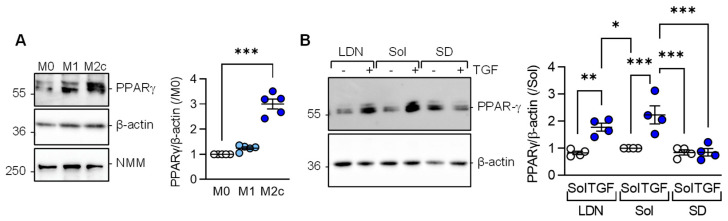
TGF-β-regulated PPAR-γ expression depends on the activation of Alk5. (**A**) Expression of PPAR-γ in M0, M1, and M2c polarized macrophages from wild-type mice; *n* = 5 independent experiments (Kruskal–Wallis test followed and Dunn’s multiple comparisons test). Non muscle myosin (NMM) was included as an additional loading control. (**B**) Expression of PPARγ in M1 polarized murine macrophages treated with solvent (0.1% DMSO), the Alk1 inhibitor; LDN193189 (100 nmol/L), or the Alk5 inhibitor; SD208 (500 nmol/L) for 2 h prior to the addition of TGF-β for M2c polarization; *n* = 4 independent experiments/group (two way ANOVA and Sidak’s multiple comparisons test). * *p* < 0.05, ** *p* < 0.01, *** *p* < 0.001.

**Figure 4 cells-12-00700-f004:**
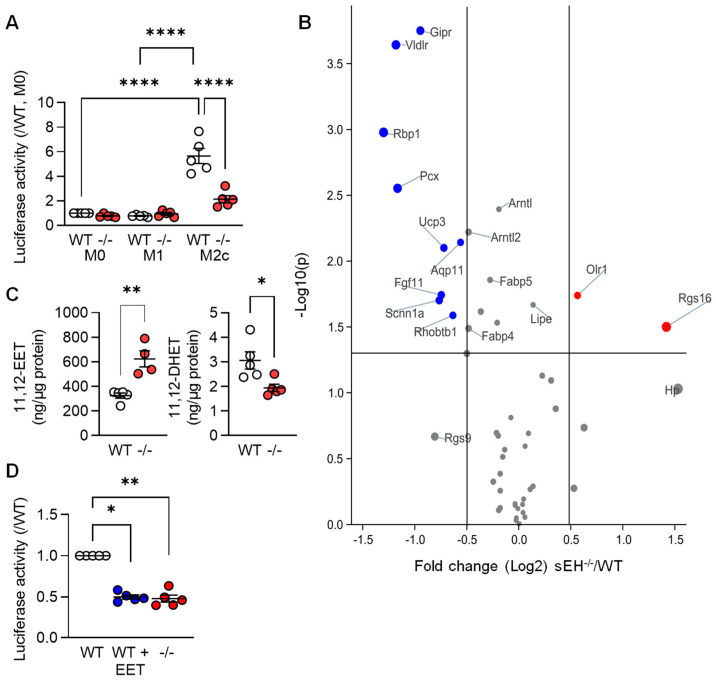
PPAR-γ activity in differentially polarized macrophages from wild-type and sEH^−/−^ mice. (**A**) Activity of a PPAR-γ-luciferase construct in M0, M1, and M2c macrophages from wild-type (WT) and sEH^−/−^ (−/−) mice; *n* = 5 independent experiments (two way ANOVA and Tukey’s multiple comparisons test). (**B**) Volcano plot showing the expression of known PPAR-γ-regulated genes in M2c macrophages from wild-type (WT) and sEH^−/−^ mice. Dataset as in Figure 1; *n* = 4 independent experiments. Blue = genes significantly downregulated and red = genes significantly upregulated in sEH^−/−^ mice versus wild-type. Grey indicates no significant alteration. (**C**) 11,12-EET and 11,12-DHET levels in M2c macrophages from wild-type and sEH^−/−^ (−/−) mice (*n* = 5 independent experiments; Student’s *t* test). (**D**) PPAR-γ activity in M2c polarized macrophages from wild-type mice treated with solvent (0.1% DMSO) or 11,12-EET (1 µmol/L, 30 min prior to TGF-β). Solvent-treated cells from sEH^−/−^ mice were included as control; *n* = 5 independent experiments (Kruskal–Wallis test followed and Dunn’s multiple comparisons test). * *p* < 0.05, ** *p* < 0.01, **** *p* < 0.0001.

**Figure 5 cells-12-00700-f005:**
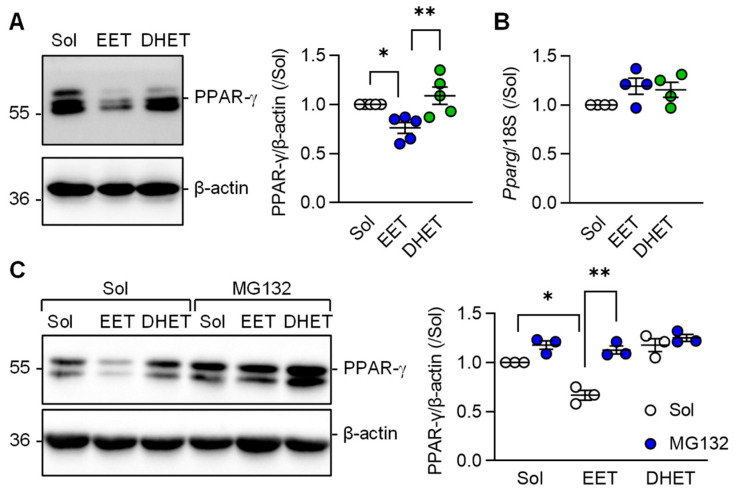
Regulation of PPAR-γ levels by 11,12-EET and 11,12-DHET in M2c polarized macrophages from wild-type mice. (**A**) Impact of solvent (Sol, 0.1% DMSO), 11,12-EET or 11,12-DHET (both 1 µmol/L; 30 min prior to TGF-β) on the expression of PPAR-γ in M2c polarized macrophages (*n* = 5 independent experiments, Kruskal–Wallis test followed and Dunn’s multiple comparisons test). (**B**) PPAR-γ mRNA levels in cells treated with solvent, 11,12-EET and 11,12-DHET as in panel a (*n* = 4 independent experiments, Kruskal–Wallis test followed and Dunn’s multiple comparisons test). (**C**) Consequence of inhibiting protein degradation using MG132 (2 µmol/L, 1 h pretreatment) on PPAR-γ protein stability in M2c polarized macrophages treated with solvent (Sol, 0.1% DMSO), 11,12-EET or 11,12-DHET (both 1 µmol/L) 1 h prior to the addition of TGF-β (*n* = 3 independent experiments, two way ANOVA and Sidak‘s multiple comparisons test). * *p* < 0.05, ** *p* < 0.01.

**Table 1 cells-12-00700-t001:** PCR primers used.

Gene	Forward	Reverse
*18S*	ctttggtcgctcgctcctc	ctgaccgggttggttttgat
*Pparg*	acaagagctgacccaatggt	tgaggcctgttgtagagctg
*Cxcr4*	atggaaccgatcagtgtgagt	tagatggtgggcaggaagatc
*Ptgs2*	gctgtacaagcagtggcaaa	ccccaaagatagcatctgga
*Ptx3*	cctgctttgtgctctctggt	tctccagcatgatgaacagc

## Data Availability

All the data are provided in the paper and its Appendix A.

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
