# Peer review of "11,12-EET Regulates PPAR-γ Expression to Modulate TGF-β-Mediated Macrophage Polarization"

_cells, 2023, doi:10.3390/cells12050700_

Round 1

Reviewer 1 Report

 The manuscript titled as “11,12-EET regulates PPAR-γ expression to modulate TGF- β -me diated macrophage polarization” dissects TGF- β induced M2 phenotype” shows the following findings:

1.      Identified the role of TGF- β receptor 1-ALK5 in PPAR-γ signaling.

2.      RNA Sequencing data showed PPAR-γ and arachidonic acid metabolism pathways are significantly upregulated in M2c compared to M1.

3.      TGF- β induced ppar-g mRNA and PPAR-γ protein level increase executes M2C function. PPAR-γ inhibition reduces phagocytic activity in M2c.

4.      sEH substrate; 11,12-EET, effectively prevented the TGF-β-induced increase in PPAR-γ protein levels in murine macrophages.

5.      Mechanism of s EH induced PPAR-γ inhibition is proteasomal degradation  of PPAR-γ.

However there are some concerns:

Major comment:

The authors mentioned:

1.      TGF- β repolarized macrophages from animals lacking the soluble epoxide hydrolase (sEH), however, responded differently and expressed lower levels of PPAR- γ -regulated genes.

Could the authors show how does the level or degree of M2C phenotype transition and/or degree of pre-resolving activity is affected in macrophages from sEH null mice compared to WT mice in presence or absence of EET supplement?

Since this study focuses on the importance of 11,12 EET on macrophage activation and resolution of inflammation, It is important to show functional aspects of M2C activity via PPAR-γ in the described experimental setting.

2.      While depicting M2C phenotype showing levels of specific murine M1 vs M2C Macrophage markers would be beneficial.

3.      It is not clear which groups have been compared and selected for western blot data as shown in supplementary original blots.

Minor comments:

1.      Please check and correct the macrophage phenotypes in lines 164, 236 etc.

2.      Please use another color for comparison between more than two groups. Always mention the color codes in the graph.

3.      Since, the time points for cellular differentiation are different for M0 M1 and M2c. As a control experiment please show a supplementary western blot for Figure 3A to check if PPAR-γ levels are altered between in M1 and M2 macrophages due to additional time of differentiation (12 hr for M1, 48 Hr for M2).

4.      Please use the same orientation for western blot and graph in Figure 3B

5.      In Figure Legend 3B Why did you compare M1 macrophages with solvent treatment to M2C macrophages with alk1/5 inhibitor? M2C controls will be important.

6.      Please mention the mutant genotype in all figures for more clarity.

7.      Please mention the mRNA in small letters in figure 5B.

8.      Please specify the time of MG132 treatment in Figure 5C.

9.      Please mention the solvents used in all the treatments.

Author Response

see attached PDF file

Reviewer 2 Report

The manuscript is generally well-written. Also, the results are convincing, and the conclusions are appropriate.

Please pay attention to typos or grammatic errors.

Author Response

See attached PDF file

Reviewer 3 Report

In this manuscript the authors showed that the differentiation from M1 to M2c macrophages by TGFß is associated with an increase in PPARgamma. Inhibition of PPARgamma reduced the phagocytic capacity of these macrophages. 11,12-epoxyeicosatrienoic acid (EET) surprisingly reduced TGF-ß-associated PPARgamma activation, an effect which might be caused by enhanced proteasomal PPARgamma degradation.

In principle this manuscript is concise and well written. However, seemingly missing repetitions of experiments, statistics that need improvement and missing control mice (cre+ no flox) stands against acceptance of the manuscript in its present form.

I suggest to encourage the authors to resubmit a revised version, where the following points have been addressed.

Major points:

Mice:

Mouse strains have not been adequately described here to unambiguously assign the effects observed here to the sEH knockout. Since the authors used wt mice as control it is important to mention the genetic background of these mice, J or N (with N having several sub-strains), since it is already well established that N and J mice show relevant genetic differences which significantly influence their performance with respect to a variety of physiologic parameters. Since the authors use the cre-flox system to knock out the sEH gene it is also important to show that ubiquitous expression (under control of the Gt(ROSA)26Sor promoter) of Cre recombinase does not affect PPARgamma induction and concentrations of 11,12-EET and 11,12-DHET.

Ideally, sEH-/- mice (which are in fact probably Crehet sEHflox/flox) should be compared to their Cre- sEHflox/flox littermates which than act as sEH-expressing controls.

Reproducibility:

The authors did not mention how often the presented experiments have been performed independently.

I think it is important that experiments from which scientific conclusions are drawn are repeated at least once, in order to exclude simple one-time mistakes, (which can happen to anyone) as actual cause for seeming effects. Some of the observations described in this manuscript have been confirmed by continuative experiments (like luciferase activity in M2c macrophages in sEH knockout mice) whereas others lack such confirmation.

Statistics:

The authors describe in th M&M section, which tests they used for statistical analysis, but not, under which conditions which test will be used. Thus, usage of tests appears somewhat arbitrary.

For example: It is unclear to me why the authors used a Two-Way ANOVA for statistical analysis in panel B of Figure 3. Have all groups been compared with all others? In the figure it appears as if only Sol and TGF had been compared with one another within each of three groups, which would require Students T-test (or corresponding non-parametric tests). Even if all groups were compared with one another, why hasn’t a One-Way ANOVA been sufficient, similar to panel A? (A corresponding question is raised regarding Figure 5).

Several datasets have been normalized so that the control group is set to “1” (RFU/cell (/WT) in Fig. 2B,  PPARg/ß-actin in Fig. 3A and B and Fig. 5 A and C, PPARg/18S in Fig. 5 B, luciferase activity in Fig. 4 A and C,).  A normalization is of course possible, but it is extremely improbable that all samples in the respective control group had the identical value which would give "1" for all probes of the control group upon normalization. An acceptable way of normalization would rather be to calculate the mean of the values in the control group and to compare also the individual values of the samples in the control group with this mean. By that, the individual samples from the control group will also distribute around the respective mean, rather than pretending an identical value for all.

This will also solve the problem that the statistical analyses presented for these experiments are not correct:

If all samples within one group have the same value it is not OK to use Student’s T test. This test requires normal distribution and equal SDs for both groups, both of which is not the case with a group that only contains identical values (= zero variance). One could use a Welch’s t-test which does not require equal variances (but just simulates a One-Sample t-Test if one group has zero variance)  – or a non-parametric test. The same is true for the comparison of more than two groups: Also an ordinary ANOVA requires Gaussian distribution of residuals and equal SDs which is definitely not the case with all values in one group being identical. Thus, in such cases Brown-Forsyth (which still requests Gaussian distribution) or non-parametric Kruskal-Wallis Test with Dunn’s multiple comparisons test can be used.

However, even if normalization of data will be performed as suggested above: Use of Student’s t-test or ANOVA should be preceded by tests for Gaussian distribution within each group and - when this test has been passed - for sufficiently equal SDs of all groups.

Regarding presentation of multiple comparisons in Figures: if all groups are compared with all others, respectively, but only asterisks are shown but not those comparisons without significant differences, this fact should be described in the Figure captions.

Experimentation:

Alk1- inhibitor

According to Fig. 3 Alk1 inhibitor LDN193189 (1 nmol/L) did not prevent a TGF-ß-induced increase in PPARgamma.

Is there any possibility to show that this inhibitor at this concentration effectively inhibits Alk1 activity?

This question is especially evident, since the concentration of the Alk5 inhibitor had been used at a 500-fold higher concentration than the Alk1 inhibitor. The concentration on 1 nM LDN193189 is rather near to a published IC50 value for its inhibition of Alk1 and might, thus, be not sufficient for full inhibition.

                Western blots

In order to enable comparison of PPARg protein between samples the authors used ß-actin as a reference for normalization. However, RNA expression data from the supplementary file indicate that upon differentiation from M0 to M1 the ACTB RNA is increased by a factor of 1.5 and drops again to the M0 level upon further differentiation to M2c. Thus, addition of TGFß to M1 macrophages causes a reduction of ACTB mRNA. If this change is reflected on ß-actin protein level, a slight apparent increase of the amount of a given protein upon addition of TGFß to M1 macrophages might in fact be caused by the reduction of the reference protein rather than an actual increase of the protein of interest. Since the changes in PPARg protein upon treatment with ALK inhibitors depicted in Figure 3 are only moderate it might be worth to also check additional reference proteins in this western blot assay to verify that the observed changes in protein amounts are real. (This might be combined with a repetition of that experiment with higher amounts of ALK1 inhibitor if - based on their experience with that compound - the author agree, that the concentration of this inhibitor might be an issue. … Or with other Alk1 inhibitors.)

Minor points:

Line 164: „pro-resolving M2a”   should be “pro-resolving M2c”

Line 175: should probably read as” transcription factors”

Line 231: “Figure 4C” should read “Figure 4B”.

 Line 233f: What does the sentence „The levels of a series of fatty acid epoxides are metabolized …” mean?

 Figure 1B: Lines should be in front of dots, since now especially the dot corresponding to PPARg cannot be localized in the plot. Why are some dots shown as empty circles (including Tnfsf15)?

 The authors write that EET is established as an activator of PPARgamma. Thus, it is not clear to me why “treating M1 polarized macrophages from wild-type mice with 11,12-EET (1 μmol/L) prior to the repolarization with TGF-ß, also decreased PPARgamma activity (Figure 4D).”  Does the initial activation with 11,12-EET cause the degradation of PPARgamma so that less activation by TGF-ß is possible? How is the situation in the sEH knockout mouse? Wouldn’t the presence of constitutively higher concentrations of PPARgamma activator EET lead to a higher PPARgamma activity (both in presence and absence of TGFß)?  The authors already commented on this in their discussion, but the interpretation is still not fully clear to me. Could the authors elaborate that discussion a bit, please?

Questions to the Authors:

Does the increased or reduced activities of PPARg in sEH knockout mice or 11,12 EET treated cells reflected in corresponding changes in phagocytosis? This would strengthen the interpretation of an impact of 11,12-EET on macrophage function.

How specific are the utilized Alk Inhibitors for the respective targets analyzed here?

Do the results of this work provide implications for therapeutic approaches?

Author Response

see attached PDF file

Reviewer 4 Report

The manuscript titled "11,12-EET regulates PPAR-g expression to modulate TGF-b-mediated macrophage polarization" by Xiaoming Li et al., explores the changes in gene expression associated with the repolarization of M1 macrophages toward macrophages with M2c phenotype influenced by TGF-beta. Although the study was carried out adequately, and technically sound, there are some points that need to be answered before publication.  

Major points:  

Introduction

The manuscript needs background information to understand the purpose of investigation, the scientific relevance is not emphasized. A brief description of the participation at the immunological level of the different phenotypes of M2 macrophages is necessary, especially since the paper focuses attention on the M2c phenotype, whose function and/or importance in various pathologies would have to be described. Why is important evaluated the influence of TGF beta in the macrophage polarization to M2c?

line 51. Expand a bit the information regarding the sources that produce TGF-beta, what cell types produce it, and under what other pathological circumstances, in addition to cancer, its production is induced. Moreover, atherosclerosis and its relationship with macrophages M2c should be addressed in the introduction or section 3.3.

Material and methods

It is necessary add details how the bioinformatics results were obtained and analyzed. This part is fundamental in the manuscript.

Results

In section 3.3. Briefly add more information to link the alteration of arachidonic acid metabolism and the development of atherosclerosis.

Discussion

 In my opinion the discussion is too long a very speculative and unfocused. I recommend reducing the discussion and focus on the results shown in the experimental report. It should be rewritten into more concise version. Moreover, authors overlook some literature about TGF-beta and its influence on macrophages polarization, two references were omitted, please include them in the introduction or discussion section.

TGF-β induces M2-like macrophage polarization via SNAIL-mediated suppression of a pro-inflammatory phenotype. Zhang F, Wang H, … Oncotarget. 2016 Aug 9;7(32):52294-52306.

MSC-secreted TGF-β regulates lipopolysaccharide-stimulated macrophage M2-like polarization via the Akt/FoxO1 pathway. Liu F, Qiu H, Xue M, …Stem Cell Res Ther. 2019 Nov 26;10(1):345.

Minor points:

In Figure 1A, the sample name should match the sample name used in the figure legend. Please, modify so that they are similar.

Lines 129-130 specify which cytokines, and how much was used so that the M0 were polarized to M1 and M2c.

There are some grammatical errors per read. This needs grammatical editing by the authors

Line 147 there is a data of the brand of the antibody that was omitted (), please add it.

Line 187, "the expression" is repeated, remove.

Line 191. Remove "etc"...it is not necessary.

Line 195. “…as well as to support the TGF-b-induced induction of a pre-resolving phenotype.”…This need grammatical editing.

Line 207. "mediated the "...should not be italicized.

Line 209. “Alk1 inhibitor” should be written in upper case, similar to ALK5 inhibitor.

Figure 4 is not described in the text in consecutive numerical order. It is not correct Figure 4A, next Figure 4C in the text.

Author Response

See attached PDF file

Round 2

Reviewer 1 Report

The manuscript can be accepted in its current form.

Author Response

We would like to thank the reviewer for his/her constructive comment and have hopefully found all of the remaining grammatical and spelling errors.

Reviewer 3 Report

The authors have responded satisfactorily to the points I have raised.

There is only one minor point left, or rather a suggestion to show the used non-muscle myosin as a reference in western blots also in the manuscript, not only in the authors' response. In principle I think the manuscript can be published.

Author Response

We have followed the reviewers suggestion and added the NMM blot to Figure 3A.

Reviewer 4 Report

The authors have taken into account the suggestions, I believe that the manuscript should be published.

Author Response

(The authors gave the same response as above.)
